# ROBUSTNESS EVALUATION USING LOCAL SUBSTITUTE NETWORKS

## ABSTRACT

Robustness of a neural network against adversarial examples is an important topic when a deep classifier is applied in safety critical use cases like health care or autonomous driving. In order to assess the robustness, practitioners use a range of different tools ranging from the adversarial attacks to exact computation of the distance to the decision boundary. We use the fact that robustness of a neural network is a local property and empirically show that computing the same metrics for the smaller local substitute networks yields good estimates of the robustness for lower cost. To construct the substitute network we develop two pruning techniques that preserve the local properties of the initial network around a given anchor point. Our experiments on MNIST dataset prove that this approach saves a significant amount of computing time and is especially beneficial for the larger models.

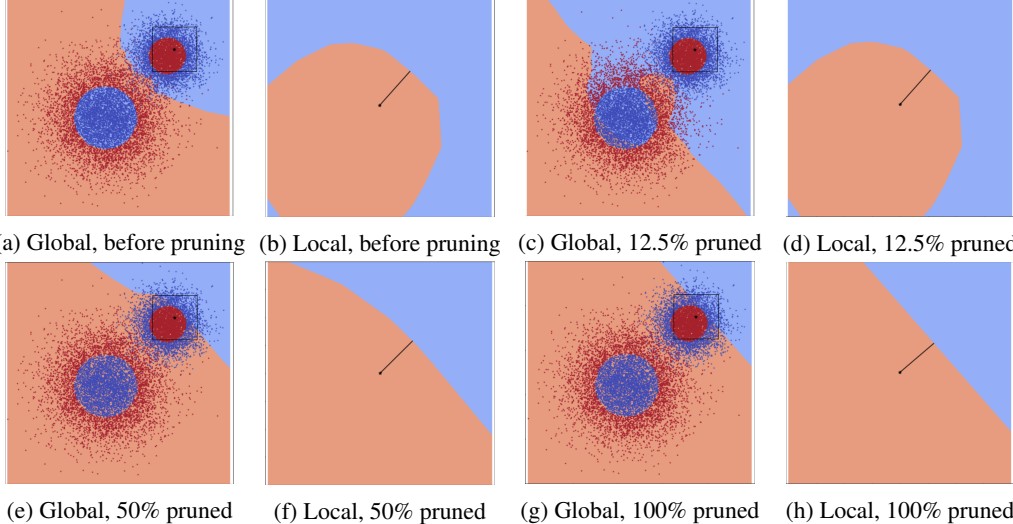

| (a) Global, before pruning | (b) Local, before pruning | (c) Global, 12.5% pruned | (d) Local, 12.5% pruned |

| (e) Global, 50% pruned | (f) Local, 50% pruned | (g) Global, 100% pruned | (h) Local, 100% pruned |

Figure 1: A toy example in the two-dimensional setting. We show the boundaries of a classifier, before (1a and 1b) and after the pruning (1c - 1h), when up to the 100% of the hidden neurons are removed. The sample we are interested in is marked by the black point and the square around it on the plots with the global view (1a, 1c, 1e and 1g) shows the local region that is depicted on the other four plots (1b, 1d, 1f and 1h). While the global behaviour changes a lot, the boundary around the chosen anchor point remains similar. Most importantly, the distance to the closest adversarial from the anchor point (shown by the black line on the plots 1b, 1d, 1f and 1h) does not change significantly. Note, that this applies even in the extreme case when we prune all the hidden layers and what remains is a linear classifier (1g and 1h). That means, in order to solve the complex task of finding the distance to the decision boundary for the initial model, we save cost by working with the simple local substitute and still get a good approximation of the exact solution.

## 1 INTRODUCTION

Impressive success of neural networks in a variety of complicated tasks makes them irreplaceable for the practitioners in spite of the known flaws. One of the problems that continuously gains more attention is the robustness of the deep neural classifiers. While multiple notions of robustness exist, depending on the use case, we consider the most basic concept of the robustness against adversarial examples - small perturbations of the correctly classified samples that lead to a false prediction. Presence of the adversarial examples severely limits the application of the neural networks in safety critical tasks like health care and autonomous driving, where the data is collected from sensors and it is not acceptable that the same image, for example a road sign, is classified differently depending on the signal noise.

While this problem is widely known, formal robustness verification methods do not allow for an assessment of the classifier's robustness when the network is large or require specific modifications in network's architecture or training procedure. In fact, Katz et al. (2017) show that the exact verification task for the ReLU classifiers is NP-complete. Therefore, constructing adversarial attacks and measuring the magnitude of the perturbation, that is required to change the prediction, is still one of the most popular ways to estimate the network's robustness. The farther away the adversarial point is from the initial sample, the more robust behaviour we expect from the network around this point. Unfortunately, the distance to an adversarial point provides only an upper bound on the distance to the decision boundary. On the other side, formal verification methods output a lower bound on that value by certifying a region around the sample as adversarial-free.

In this work we develop a novel inexact robustness assessment method that utilizes both techniques as well as the fact, that robustness of a network against adversarial perturbations around a given sample is a local property. That is, it does not depend on the behaviour of the network outside of the sample's neighborhood. In other words, two networks with the similar decision boundaries around the same anchor point must have similar robustness properties, despite showing completely different behaviour away from its local neighborhood. Based on these observations we

1. **develop a novel method to assess the robustness of the deep neural classifiers** based on the local nature of the robustness properties,

2. **develop two pruning techniques** that remove the non-linear activation functions to reduce the complexity of the verification task, but preserve the local behaviour of the network as much as possible: one that is based on a bound propagation technique and replaces specific activations by a constant value, one that preserves the output of the initial network for one adversarial point and replaces activation functions by the linear ones,

3. **empirically verify that the robustness metrics computed on the pruned substitute networks are good estimates of the robustness of the initial network** by conducting experiments on the MNIST dataset and convolutional networks of different sizes.

On Figure 1 we show an example of the difference in the decision boundary before and after we apply one of the proposed pruning techniques in the two-dimensional setting. We remove up to all of the hidden neurons while retaining the important properties of the decision boundary locally around the base point.

This work is organized as follows. In Section 2 we introduce the necessary notation and formalize the context of our analysis. In Section 3 we develop both pruning techniques and explain, how do we put the focus on the local neighborhood around the base sample. Further, in Section 4 we set up the experimentation workflow and show the results. In Section 5 we mention the relevant related work and, finally, we draw the conclusions including options for the future research in Section 6.

## 2 NOTATION

The general idea as well as our pruning methods are applicable to any type of deep classifier and the constraints apply depending on the deployed attack, verification and bound propagation approaches. However, in order to allow for a simpler comparison with the existing attacks and verification techniques we develop our analysis for the classification networks that Li et al. (2023) use for their comprehensive overview and toolbox of robustness verification approaches.

We consider a neural network $f$ that consists of $L$ linear layers, dense or convolutional, and $L-1$ activation layers. We denote the number of neurons in each layer $l$ as $n_l$ for $l \in \{0, \ldots, L\}$, where the zero's layer contains the input neurons. Furthermore, we consider the $L^\infty$-norm as the perturbation norm since it is supported by the majority of the verification methods. For a correctly classified anchor point $x^0 \in \mathbb{R}^{n_0}$, weight matrices $W^l \in \mathbb{R}^{n_l \times n_{l-1}}$, and bias vectors $b^l \in \mathbb{R}^{n_l}$, the output of the $i$-th neuron in the $l$-th layer before applying the activation function $\sigma$ is $f_i^l(x^0) = W_{i,:}^l x^{l-1} + b_i^l$ and for the last layer $f^L(x^0) = W^L x^{L-1} + b^L = f(x^0)$. Next, we introduce the notation for four metrics that are commonly used to assess the robustness of the given classifier. All these algorithms take the network's parameters $\{W^l, b^l\}_{l \in \{1, \ldots, L\}}$ and $x^0$ as input.

**Pre-activation bounds propagation**   A pre-activation bounds propagation (of just bounds propagation) algorithm $\mathcal{B}$ is an algorithm that additionally takes a perturbation budget $\epsilon$ as an input and returns a set of the lower and upper bounds $\{\underline{a}^l, \bar{a}^l\}_{l \in \{1, \ldots, L\}}$ on the possible pre-activation values $W^l x^{l-1} + b^l$. In other words, for all possible points $\tilde{x}^0$ with $\|x^0 - \tilde{x}^0\|_\infty \leq \epsilon$ it must hold that $\underline{a}^l \leq W^l x^{l-1} + b^l \leq \bar{a}^l$ in every layer $l$.

**Adversarial attack**   An adversarial attack algorithm $\mathcal{A}$ is an algorithm that returns a point $x_{\text{adv}}^0$ that is assigned by $f$ to a different class than $x^0$. We denote the magnitude of the adversarial perturbation $\|x^0 - x_{\text{adv}}^0\|_\infty$ by $\mathcal{A}(f) = d_{\text{adv}}$. The smaller the distance $d_{\text{adv}}$, the better.

**Robustness verification (incomplete)**   A robustness verification algorithm $\mathcal{R}$ is an algorithm that for a given radius $\epsilon$ either returns a certificate, proving that the neighborhood set $\mathbb{B}_\epsilon(x^0) = \{x \mid \|x^0 - x\|_\infty \leq \epsilon\}$ does not contain any adversarial points (that is all points inside get the same label by $f$), or abstains. The latter might happen either because $\mathbb{B}_\epsilon(x^0)$ does indeed contain an adversarial point or due to the incompleteness of $\mathcal{R}$. We denote the largest $\epsilon$ such that $\mathcal{R}$ outputs the robustness certificate for the corresponding $\mathbb{B}_\epsilon(x^0)$ as $\mathcal{R}(f) = d_{\text{rob}}$.

**Distance to the decision boundary**   Finally, an algorithm $\mathcal{D}$ computing the distance to the decision boundary from $x^0$ is an algorithm that outputs the distance $\|x^0 - \tilde{x}_{\text{adv}}^0\|_\infty$ to the closest adversarial point $\tilde{x}_{\text{adv}}^0$. Formally, $\mathcal{D}$ must solve the following optimization problem.

$$\min_{\tilde{y} \neq y} \min_{\tilde{x} \in \mathbb{R}^{n_0}} \|x^0 - \tilde{x}^0\|_\infty \text{ s.t. } f_y(x^0) \leq f_{\tilde{y}}(\tilde{x}^0), \tag{1}$$

where $y$ is the correctly predicted label of $x^0$. We denote the optimal objective function value of the minimization problem 1 by $\mathcal{D}(f) = d_{\text{bnd}}$.

Note, that for any $\mathcal{A}$, $\mathcal{R}$ and $\mathcal{D}$ it holds that $d_{\text{rob}} \leq d_{\text{bnd}} \leq d_{\text{adv}}$. When we consider a pair of networks $f$ and $g$ we define $\hat{d}_{\text{adv}}$, $\hat{d}_{\text{rob}}$ and $\hat{d}_{\text{bnd}}$ as $\mathcal{A}(g)$, $\mathcal{R}(g)$ and $\mathcal{D}(g)$ correspondingly.

## 3   LOCAL SUBSTITUTES

Given the initial deep classifier, in order to construct a neural network that allows for the efficient robustness verification we have to effectively reduce its size. Practitioners pursue the same goal when the network has to be deployed in a setting with strictly constrained available memory or when the forward pass during inference must be particularly fast. In these scenarios it is common to prune the network by removing its neurons and weights, while keeping the drop in performance on the main task as low as possible and even improve generalization to the new data.

However, these available pruning techniques are not suitable for our application. First, all pruning techniques act globally and take into account the behaviour of the network on the whole available dataset. Instead, we show how to prune the network such that only the vicinity of a particular point is taken into account that determines its robustness properties. Second, when we assess the robustness of a network that is already trained, we do not care about the classification error of the resulting network. The most important goal for us, after reducing the network complexity, is to preserve a similar decision boundary in a small region around the anchor point. For more information about the available pruning techniques see Section 5.

To achieve the primary goal - a faster verification, we have to reduce the number of non-linear transformations within the neural network. The non-linear nature of the neural networks and the large

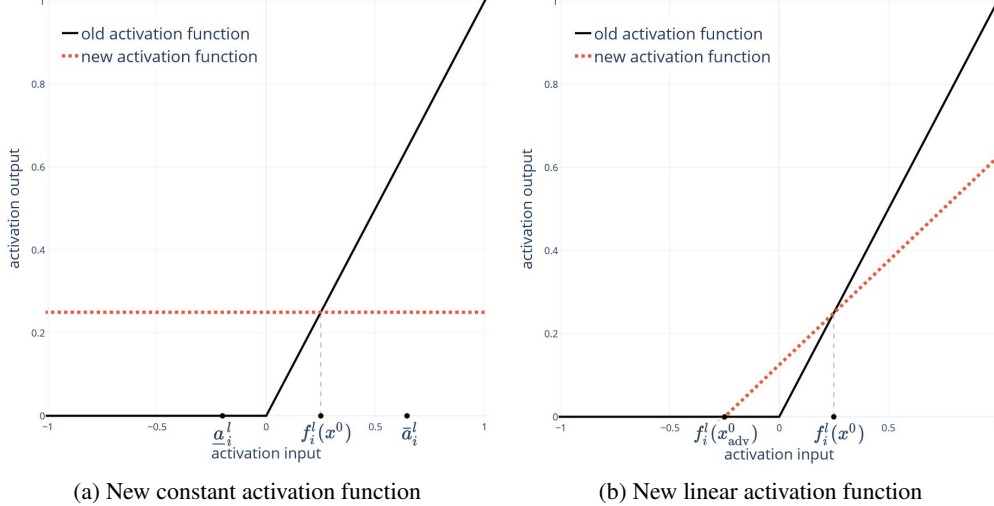

(a) New constant activation function  (b) New linear activation function

Figure 2: Two options we use to eliminate the non-linear activation function and the corresponding neuron. On the left, Figure 2a shows the constant function that always outputs the value $f_i^l(x^0)$. On the right, Figure 2b shows the linear function that is defined by preserving the neuron's output for two points $x^0$ and $x^0_{\text{adv}}$.

number of the non-linear activation functions in its hidden layers are the main reason why the deep networks are able to perform well on complex, high-dimensional data by learning meaningful representations. On the other hand these non-linear transformations make it hard to explain and verify other properties of the network like robustness. While the exact reasons for that vary depending on the verification approach, a common problem is to formally describe the output sets of the points from the neighborhood $\mathbb{B}_\epsilon(x^0)$. The more non-linear transformation does this set undergo, the more complex becomes the output and the worse are the possible approximations. For our task, we remove the non-linear functions from the network by replacing them by the linear transformations and removing the corresponding neurons from the network.

### 3.1 BOUNDS PROPAGATION BASED ITERATIVE PRUNING

Bounds propagation techniques, as defined in Section 2, are widely used in verification literature. The most simple way to utilize them for the ReLU networks is to detect the neurons that does not change their state for the inputs from $\mathbb{B}_\epsilon(x^0)$. That is, if $\bar{a}_i^l \leq 0$ the input to the neuron $i$ from layer $l$ is always negative or zero and the neuron remains deactivated such that the activation function returns zero for all relevant inputs. Also, if we get the case $\underline{a}_i^l \geq 0$, this neuron is never deactivated and effectively acts as the identity function. By pruning these stable neurons, we get the new smaller network $g$ with exactly the same output $g(\tilde{x}^0) = f(\tilde{x}^0)$ for all $\tilde{x}^0 \in \mathbb{B}_\epsilon(x^0)$.

Unfortunately, this does not work for the not piece-wise linear activation functions like sigmoid. Moreover, in this procedure the only parameter that controls the number of the removed non-linear transformations is $\epsilon$ that defines the size of $\mathbb{B}_\epsilon(x^0)$. The smaller the input region around $x^0$, the tighter are the bounds $\underline{a}_i^l, \bar{a}_i^l$, such that less of the neurons remain in the undecided state $\underline{a}_i^l \leq 0 \leq \bar{a}_i^l$. However, we use this parameter to specify the region that we take into account for pruning. When $\epsilon$ is chosen too small to remove more neurons, we risk that the decision boundary around $x^0$ will lie completely outside of $\mathbb{B}_\epsilon(x^0)$ meaning that the whole set will be assigned to the same class as $x^0$. This is not desirable since the computation of the bounds in $\mathcal{B}$ does not account for the behaviour of the network at the points outside of the given $\epsilon$-neighborhood around $x^0$, which, in this case, does determine the network's robustness properties. That means we would miss essential information about the network when computing the bounds $\underline{a}_i^l, \bar{a}_i^l$ in the first place.

Instead, we first set $\epsilon = d_{\text{adv}}$ by running an attack $\mathcal{A}$. This way we ensure, that $\mathbb{B}_\epsilon(x^0)$ is not too small and provably contains the decision boundary between the points with different predicted classes $x^0$ and $x^0_{\text{adv}}$ (both are points from $\mathbb{B}_\epsilon(x^0)$). Second, given a user defined ratio $\gamma \in (0, 1)$ of

the neurons to be pruned, we apply the bounds propagation algorithm $\mathcal{B}$ and remove the neurons with the smallest difference $\bar{a}_i^l - \underline{a}_i^l$, that we use as a measure of the neuron's variability given the input perturbation budget $d_{\text{adv}}$. That is, we prune the neurons with the least variable output, considered the inputs from $\mathbb{B}_{d_{\text{adv}}}(\boldsymbol{x}^0)$. In order to actually remove the neuron, we replace the activation function by the constant value $f_i^l(\boldsymbol{x}^0)$ that this neuron outputs at $\boldsymbol{x}^0$. Figure 2a shows how the activation function changes on the example of a ReLU neuron. Finally, we observe that the neurons of the first layer have $\bar{a}_i^1$ and $\underline{a}_i^1$ with the smallest gap due to the magnification during bounds estimation when $\bar{a}_i^l$ and $\underline{a}_i^l$ are propagated in $\mathcal{B}$ and become looser. Therefore, to prevent the elimination of the neurons exclusively from the first layers, we apply the pruning iteratively by removing a small portion at a time and then recomputing the pre-activation bounds. This way, each time we prune neurons with the most reliable bounds and distribute the removed neurons more evenly throughout the layers. We present the pseudo code for the described method in Algorithm 1.

---

**Algorithm 1:** Bounds propagation based iterative pruning

---

1: **Input:** Algorithms $\mathcal{A}$, $\mathcal{B}$, network $f$, point $\boldsymbol{x} \in \mathbb{R}^{n_0}$, $\gamma \in (0, 1)$ and $\gamma_0 \ll \gamma$.
2: **Output:** Network with at most $(1 - \gamma)(n_1 + \cdots + n_L)$ neurons.
3: Compute $d_{\text{adv}} = \mathcal{A}(f)$.
4: Compute $\bar{a}^l$, $\underline{a}^l$ using $\mathcal{B}$ with $\epsilon = d_{\text{adv}}$, initialize $k = 1$.
5: **repeat**
6:     Compute the threshold $\delta$ as the $\gamma_0$-quantile of the values $\bar{a}_i^l - \underline{a}_i^l$ for $l \in \{2, \ldots, L\}$ and
    $i \in \{1, \ldots, n_l\}$.
7:     **for** $l = 2, \ldots, L$ **do**
8:         Define $I^l \in \{\text{True}, \text{False}\}^{n_l}$ as the indicator vector of the remaining neurons:
        $I_i^l = \mathbb{1}_{\bar{a}_i^l - \underline{a}_i^l > \delta}$, and $J^l$ as $J^l = \neg I^l$
9:         Update the current layer: $\boldsymbol{W}^l = \boldsymbol{W}_{I^l,:}^l$ and $\boldsymbol{b}^l = \boldsymbol{b}_{I^l}^l$
10:         **if** $l < L$ **then**
11:             Update the next layer: $\boldsymbol{W}^{l+1} = \boldsymbol{W}_{:,I^l}^{l+1}$ and $\boldsymbol{b}^{l+1} = \boldsymbol{b}^{l+1} + \boldsymbol{W}_{:,J^l}^{l+1} f(\boldsymbol{x}^0)_{J^l}$, where we
            update $\boldsymbol{b}^{l+1}$ in order to account for the non-zero output value of the removed neurons.
12:         **end if**
13:         Update $k = k + 1$.
14:     **end for**
15: **until** $(1 - \gamma_0)^k \geq 1 - \gamma$
16: **Return:** Network with the new weights $\boldsymbol{W}^l, \boldsymbol{b}^l$

---

By applying this pruning method we achieve the following. 1. **We preserve local behaviour** of the network around $\boldsymbol{x}^0$ by setting the output of the least uncertain neurons to a constant, while we don't take into account the points outside of $\mathbb{B}_{d_{\text{adv}}}(\boldsymbol{x}^0)$. 2. **We preserve the value of the network at $\boldsymbol{x}^0$**, since $g(\boldsymbol{x}^0) = f(\boldsymbol{x}^0)$ by construction of $g$, maintaining it's predicted label. 3. We have a **dedicated parameter $\gamma$ to set the desired amount of the pruned neurons**.

## 3.2 Adversarial preserving layerwise pruning

One problem that seems to be unavoidable for any type of pruning is what we call boundary collapse. When we prune too much, for example when $\gamma$ is chosen too large for the pruner described in Section 3.1, the resulting network $g$ becomes not expressive enough to maintain a non-trivial boundary even locally around $\boldsymbol{x}^0$. In this case the prediction region grows and the classifier outputs the same label for every point becoming useless for the robustness assessment of the initial network. However, we develop a simple way to solve this problem and to ensure, that even under severe simplification of the decision boundary, it does not collapse, but remains in the vicinity of $\boldsymbol{x}^0$.

Again, as for the bounds propagation based pruner, we use an adversarial attack $\mathcal{A}$ to determine the radius $d_{\text{adv}}$ of the relevant neighborhood around $\boldsymbol{x}^0$. Now, we do not only ensure that the network's output at the base point $\boldsymbol{x}^0$ remains the same, but also at the point $\boldsymbol{x}_{\text{adv}}^0$ that is assigned to a different class. That means, we force $g$ to output the same value as $f$ in the center of $\mathbb{B}_{d_{\text{adv}}}(\boldsymbol{x}^0)$ and at one point on its boundary. Therefore, $g$ must assign these points to the different classes and have the decision boundary between them like $f$.

To enforce this adversarial preserving property we replace the non-linear activation function by a linear function that has the same output for the points $\boldsymbol{x}^0$ and $\boldsymbol{x}^0_{\text{adv}}$. Figure 2b shows an example of the new activation function for a ReLU neuron. Note, that there is always exactly one linear function that satisfies this property and goes through the points $(f^l_i(\boldsymbol{x}^0), \sigma(f^l_i(\boldsymbol{x}^0)))$ and $(f^l_i(\boldsymbol{x}^0_{\text{adv}}), \sigma(f^l_i(\boldsymbol{x}^0_{\text{adv}})))$ unless $f^l_i(\boldsymbol{x}^0) = f^l_i(\boldsymbol{x}^0_{\text{adv}})$, in which case we set the new activation function to be constant as we do it above for the bounds propagation based pruner. Unlike before, we do not choose the neurons to prune according to any criterion, but instead just prune an entire layer or a set of layers specified by the user. While pruning individual neurons is possible, it leads to a network with skip connections, that are not supported by many verification algorithms. We summarize the adversarial preserving pruning procedure in Algorithm 2.

---

**Algorithm 2:** Adversarial preserving pruning

1: **Input:** Algorithms $\mathcal{A}$, network $f$, point $\boldsymbol{x} \in \mathbb{R}^{n_0}$, layer $l$ to prune (multiple layers are pruned one by one).
2: **Output:** Network without the initial layer $l$.
3: Compute $d_{\text{adv}} = \mathcal{A}(f)$.
4: Compute the coefficients $\boldsymbol{c}, \boldsymbol{d} \in \mathbb{R}^{n_l}$ such that

$$\boldsymbol{c}_i f^l_i(\boldsymbol{x}^0) + \boldsymbol{d}_i = \sigma(f^l_i(\boldsymbol{x}^0)) \text{ and } \boldsymbol{c}_i f^l_i(\boldsymbol{x}^0_{\text{adv}}) + \boldsymbol{d}_i = \sigma(f^l_i(\boldsymbol{x}^0_{\text{adv}}))$$

for all $i \in \{1, \ldots, n_l\}$. In the only case of multiple solutions, when $f^l_i(\boldsymbol{x}^0) = f^l_i(\boldsymbol{x}^0_{\text{adv}})$, define $\boldsymbol{c}_i = 0$ and $\boldsymbol{d}_i = f^l_i(\boldsymbol{x}^0)$.
5: Replace the linear transformations $l$ and $l+1$ (as well as the activation layer) by the following single linear layer: $\boldsymbol{W}^l = \boldsymbol{W}^{l+1}\text{diag}(\boldsymbol{c})\boldsymbol{W}^l$ and $\boldsymbol{b}^l = \boldsymbol{W}^{l+1}\text{diag}(\boldsymbol{c})\boldsymbol{b}^l + \boldsymbol{W}^{l+1}\boldsymbol{d} + \boldsymbol{b}^{l+1}$.
6: **Return:** Network with the new weights $\boldsymbol{W}^l, \boldsymbol{b}^l$

---

This adversarial preserving pruning method has the following advantages. 1. We ensure that $g$ outputs the **same value at $\boldsymbol{x}^0$ and, additionally, at $\boldsymbol{x}^0_{\text{adv}}$ at the boundary of** $\mathbb{B}_{d_{\text{adv}}}(\boldsymbol{x}^0)$. 2. By maintaining $g$'s predicted label at these points we **implicitly enforce that the decision boundary does not move farther away than $\boldsymbol{x}^0_{\text{adv}}$.**

## 4 EXPERIMENTS

**Setup**   We conduct the experiments using five convolutional networks with ReLU activations publicly provided by Li et al. (2023) that are pretrained on the MNIST dataset (LeCun, 1998). We call them according to authors' notation A–E, where A is the smallest network with 8000 parameters and E the largest with 600000 parameters. We evaluate the training time of all approaches on a machine with a single NVIDIA GTX 1080 Ti GPU with 11 GB memory. For the algorithms $\mathcal{A}$, $\mathcal{B}$, $\mathcal{R}$ and $\mathcal{D}$ we use the following popular approaches correspondingly (all available within the unified toolbox by Li et al. (2023)): projected gradient descent (PGD) attack, FastLinIBP interval bound propagation (combination of the methods by Weng et al. (2018) and Gowal et al. (2018)), dual verification approach based on the linear relaxations of ReLU by Wong & Kolter (2018), MILP-based exact verification by Tjeng et al. (2018). All third-party methods are used with the default parameters by Li et al. (2023).

**Procedure**   The goal is to get a good estimate of $d_{\text{bnd}}$ or $d_{\text{rob}}$ without running $\mathcal{D}$ and $\mathcal{R}$ on the full network. Therefore, for each network we apply both pruning methods (Sections 3.1 and 3.2) with varying magnitude. For our bounds propagation based pruner we use the parameters $\gamma \in \{0.5, 0.75, 0.9, 0.95\}$. We do not prune the neurons from the first layer, as it is reported to severely constraint the resulting network by other works (see Section 5). For the adversarial preserving layerwise pruning we prune an increasing number of layers starting with the second one.

Before and after the pruning, (due to the long runtime of the exact verification) we run $\mathcal{D}$ on the small models A, B and $\mathcal{R}$ on the models C, D, E on 100 samples, randomly chosen from the training set. For all the pruned networks we compute the mean absolute difference $|\hat{d}_{\text{bnd}} - d_{\text{bnd}}|/d_{\text{bnd}}$ or $|\hat{d}_{\text{rob}} - d_{\text{rob}}|/d_{\text{rob}}$ depending on the verification method used on the corresponding model. Since the fastest method to assess the robustness of a network is to perform an adversarial attack, we use

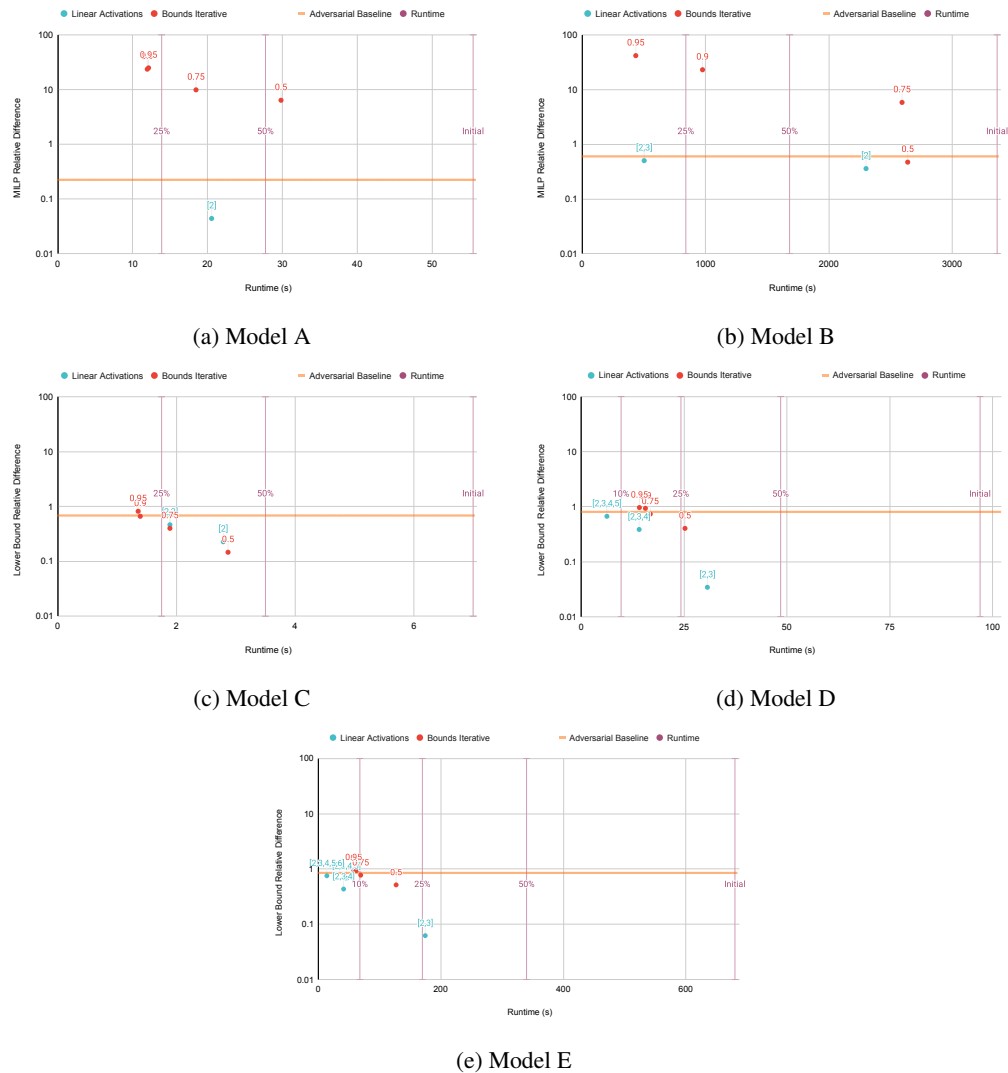

Figure 3: Experiment results showing the average absolute relative difference between the robustness metrics computed on the pruned and unpruned model.

$|d_{\text{adv}} - d_{\text{bnd}}|/d_{\text{bnd}}$ and $|d_{\text{adv}} - d_{\text{rob}}|/d_{\text{rob}}$ as the baseline. That means, the goal is to construct an estimate that is closer to $d_{\text{rob}}$ and $d_{\text{bnd}}$ than $d_{\text{adv}}$, but to do it faster than $\mathcal{R}$ and $\mathcal{D}$ when applied on the initial model.

**Results** Figure 3 shows the results for each of the models. Each point on the plot represents an evaluation of one pruned network. The $x$-coordinate is the average runtime of the verification procedure including pruning. Vertical lines represent the runtime of the same algorithm (without pruning) on the initial network as well as 50%, 25% and 10% speed-up marks. The $y$-coordinate is the average absolute relative difference in verification metrics before and after the pruning as described above. Numbers at each of the points show the amount of pruning applied to the model ($\gamma$ for the IBP-based pruner and the numbers of the removed layers by the adversarial preserving pruner). Finally, we show the baseline performance achieved by $\mathcal{A}$ by the solid horizontal line. That means a good estimator would result in a point lying in the bottom left part of the plot.

Results clearly show, that the proposed method achieve a consistent speedup of the verification procedure by at least a factor of two. Depending on the pruning magnitude, even faster verification is possible with less than 10% of the initial runtime.

Looking at the difference of the robustness verification metrics before and after the pruning we report that it improves with the size of the initial network. IBP-based pruning does not lead to any useful approximations of $d_{\text{bnd}}$ for models A and B as the relative difference is 10 or higher, while the baseline result being less than 1. However, it significantly improves for the models C, D and E. There, IBP-based pruner outperforms the adversarial baseline, except for the large values of $\gamma$ where the results are similar to the baseline. Overall, for the larger networks this pruner achieves at least twice the speed-up when used with $\gamma \in (0.5, 0.7)$ while outperforming the baseline.

The adversarial preserving pruner yields even better results. Already by removing one or two layers, it achieves a significantly lower average runtime of $50\% - 25\%$ of the runtime on the full model. At the same time, $\mathcal{B}$ and $\mathcal{D}$, when applied to the pruned networks, return good estimates of the robustness of the initial models. The relative difference in $d_{\text{rob}}$ and $d_{\text{bnd}}$ is around 0.1 when we remove less than three layers. This pruner consistently outperforms the baseline, even when the most of the neurons are pruned, as its decision boundary is always closer to the base point $\boldsymbol{x}^0$ than the adversarial point $\boldsymbol{x}^0_{\text{adv}}$ by construction (see Section 3.2). Overall, the adversarial preserving pruner provides an excellent tool for the practitioners who is not interested in the exact verification of their network, but need a faster way to estimate the robustness better than the PGD attack.

## 5 RELATED WORK

When talking about robustness verification, there are two main families of methods used in practice: complete (or exact) and incomplete methods. **Complete verification** techniques verify exactly whether an adversarial example exists inside the given perturbation budget. The main theoretical formulation of this direction uses the mixed integer linear programming (MILP) framework, for which multiple solvers are available based on either satisfiability modulo theory (SMT) Katz et al. (2017) or integer programming methods (Tjeng & Tedrake, 2017; Dutta et al., 2017).

**Incomplete verification** techniques relax the constraints that are put on the activations of each layer in the neural network. An example is the work by Wong & Kolter (2018), where the authors replace ReLU activations with a convex substitute, and, to further increase the efficiency of the method, use the dual formulation of the obtained relaxed linear programming (LP) problem. Ehlers (2017) use a similar approximation of the ReLU. However, they do not optimize over the dual LP and solve the problem using SMT. Following this direction Weng et al. (2018); Zhang et al. (2018) approximate the ReLU activation functions and propagate the possible bounds of the input to the final layer. Other approaches use semi-definite programming (Raghunathan et al., 2018a;b), Lipschitz constant (Weng et al., 2018; Hein & Andriushchenko, 2017; Zhang et al., 2019; Tsuzuku et al., 2018) and abstraction of the network (Gehr et al., 2018). Similar to our work, Croce et al. (2019) look at the local area around the base point and in particular utilize the fact that (in case of ReLU activations) it consists of regions, where the network act as a linear function. Their verification method $\mathcal{R}$ utilizes this fact and the authors additionally propose a robust training loss.

All incomplete verification approaches provide a lower bound on the distance to the decision boundary, which is strongly dependant on the tightness of the used relaxation and the way each algorithm computes the bounds on the pre-activation values in each layer. Li et al. (2023) provide a detailed overview of the verification methods including a proposed taxonomy as well as a publicly available verification toolbox.

Pruning methods allow for minimizing the size of the models at only a small cost of their accuracy. They eliminate weights, neurons or layers, such that the output changes as little as possible. Hoefler et al. (2021) divide the pruning methods into two categories: **data independent pruning** and **data dependent pruning**. The first group consists of pruning methods that does not explicitly use the training data. These methods consider the weights and nodes of the network after training and heuristically find a maximal subset of them that have the minimal impact on the final accuracy (Li et al., 2016; Han et al., 2015; Jonathan & Michael, 2019). Shumailov et al. (2019) consider the transferability of adversarial examples between the pruned (Han et al., 2015) and initial model. Their findings are connected to our work, since we also deal with the transferability of the robustness properties to the network after pruning. However, the authors suggest that adversarial examples are interchangeable only at the low levels of sparsity, which unfortunately limits the potential gain in computational cost.

**Data dependent pruning** methods consider the effect of the training data on the network's output, when making the decision of which neurons or weights to prune. Specifically, these methods take into consideration the influence of the training data on either the output, the activations or the derivatives of the network and eliminate the extraneous parts of the network, without using its weights and biases directly. This idea do not apply well to our scenario, as the training data can not be used for pruning while preserving the decision boundary locally. We mention them, as they are successful in accuracy-preserving pruning (Lee et al., 2019; Wang et al., 2020; Evci et al., 2020).

Finally, another method to create a small network with the similar robustness properties as the initial large network, is to **train it from scratch**. For example, using the initial network's output to guide the training process. Papernot et al. (2016) use this approach to create a substitute network for attacking the initial model in a black-box environment (that means without the access to its gradients). They specifically train the substitute to mimic the initial network on carefully selected data points, iteratively picked by going towards the decision boundary using the gradient information of the substitute network. This method seems very promising, as they also look at the problem in terms of creating a network with a similar decision boundary as the initial one. However, we abstain from performing training in our use case since we must do it for each of the considered base points (unlike Papernot et al. (2016), who train a single substitute network) that is potentially even slower than the verification on $f$.

## 6 CONCLUSION

We develop a novel framework to estimate the robustness of a deep classifier around a given base point. Using the available methods for the exact and inexact robustness verification we apply these to a specifically constructed local substitute network. We observe, that the decision boundary of the initial network far away from the base point does not affect its robustness against adversarial perturbations in the local neighborhood, and develop two pruning techniques that preserve local properties of the network while reducing its overall complexity. This idea allows for a more efficient application of the verification methods $\mathcal{B}$ and $\mathcal{D}$ (on the pruned models) while the resulting metrics are good estimates for the robustness of the initial network because of the similarity in the local behavior. We conduct the experiments on MNIST dataset and a variety of convolutional networks. The results show that in particular the adversarial preserving pruner (see Section 3.2) achieves good approximation of $d_{\mathrm{rob}}$ and $d_{\mathrm{bnd}}$ (around 10% relative difference, when one or two layers are removed) with a significant speed-up compared to $\mathcal{B}$ and $\mathcal{D}$ applied on the initial model. In summary, the proposed technique allows the practitioners who is interested in faster inexact robustness assessment to get a better evaluation of the robustness properties then provided by an adversarial attack.

**Future work** The success of the presented approach indicates an intriguing direction for the future work towards larger networks and datasets. Also, conducting the experiments with different $\mathcal{R}$ and $\mathcal{D}$ should verify that our approach is agnostic to the applied verification methods. Furthermore, we think about combining both pruning techniques from Section 3 by selectively pruning the least variable neurons and at the same time retain the adversarial point $x_{\mathrm{adv}}^0$. The resulting pruning approach should have the merits of the both proposed methods.

### REPRODUCIBILITY STATEMENT

We clearly and thoroughly describe the developed pruning strategies in Section 3. See Algorithms 1 and 2 for the detailed descriptions. The datasets and networks that we use are publicly available and referenced in the main text. The same applies to the attack and verification algorithms we deploy. We will provide a link to the repository that contains the project code.

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
