# OpenReview forum: "Robustness Evaluation Using Local Substitute Networks"
_ICLR.cc/2023/Conference — Submitted to ICLR 2023_

### Official Review · Reviewer_zHRW · 2022-10-19

**Confidence:** 5
**Correctness:** 2
**Technical Novelty And Significance:** 4
**Empirical Novelty And Significance:** 2
**Recommendation:** 3

**Clarity, Quality, Novelty And Reproducibility:**

The work is to the best of my knowledge novel as I haven't seen before the use of surrogate models for evaluation of the robustness.
The method section of the paper is quite clearly written and it should be feasible to re-implement the method for people reading the paper. The initial figure, combined with the abstract, gives a good intuition to what the main idea of the paper is.

On the other hand, I think that the clarity of the experiment section could be improved. There is no description of how the models have been trained. Are they the results of adversarial training, of some certified training, or are they just normally trained with cross-entropy? These are important information to help put the results into context, but all that is known is that the authors used these pre-trained models from another paper, without describing them.

**Strength And Weaknesses:**

I identify two main weakness of the paper.

# Lack of justification for the approach.
If I use a complete verification method, like a MIP solver, I can get an exact measurement of the robustness of my model around a given point. If I use an adversarial attack evaluation, I get an upper bound on the robustness of my model around a given point, telling me that my model is definitely not robust beyond a given point. If I use an incomplete verification method based on bounding, I get a lower bound on the robustness of my model, telling me that as long as perturbation are smaller than a certain amount, I'm safe.
On the other hand, if I use this method, the result that I get after running the method is.... a number. Is it an upper bound, a lower bound? We don't know. At that point, I'm uncertain as to how this information can be used. The authors claims at the end of Section 4. that it's "A faster way to estimate the robustness better than the PGD attack" but there is no real argument for why this is the case.

# Problem with the evaluation and experimental results.
- I feel like returning the information as a mean absolute difference obfuscate a lot of the details. How often is the result an overapproximation vs. an under-approximation? Are there outliers?
- For the method that is based on linear activations, the results are going to be dependent on what is the adversarial example found by the  PGD attack. How do the results vary if you rerun the method several times?
- The adversarial baseline is marked in all the results as a straight horizontal line. However, it presumably also has a specific runtime and present a valid estimate of the robustness, so it would be meaningful to represent it as a point. (I assume that the adversarial baseline is actually significantly faster than the whole compression + MIP or compression + bound).
- Any dots on the graph that is above the level of 1 is essentially vacuous. If you returned as compressed network a network that is constant and always return the wrong class everywhere, you get a $\hat{d}_bnd$ or a $\hat{d}_rob$ that is equal to 0, leading to a relative difference of 1. On Model A and Model B, we can therefore observe that the Bounds Iterative method should definitely not be used. This is problematic because this corresponds to the model where the reference is a valid measurement (the exact, closest adversarial example.)
- The evaluation of Model C, D and E on the other hand has a problem because here, the reference point is the bound computed on the full model. This is an overapproximation of the true robustness. As it is, it makes the adversarial baseline look worse. You could imagine the case where the adversarial baseline would find the true closest adversarial example, but would then get a poor score due to being far away from the bound.

It is also problematic that all evaluations is only performed on MNIST, which is known to be quite a particular dataset.



**Summary Of The Paper:**

This paper studies way of measuring the robustness of neural networks.
The goal of the paper is to provide a method to assess the local robustness of the paper at a lower cost than existing methods.
The suggested method consists in creating another, pruned neural network, that is much smaller but should be a good approximation of the original network, even if the approximation is only valid in a small area around a given point. Applying robustness measurements techniques to the smaller network is then faster than applying them to the large original network.

Two different strategies are proposed. Either a proportion of neurons are pruned in each layer, prioritizing the ones that have the smallest gap between lower bound and upper bound (because they have the least amount of variability) and replacing them by constant values. This  pruning strategy is designed such that the prediction at the original point $x_0$ does not get affected.
The other strategy consists in fusing two layers (and eliminating the non-linearity in between them). This is based by finding the linear replacement that will maintain evaluation at both the original point $x_0$ and a found adversarial example $x_\text{adv}$.

Experiments are performed on several convolutional network performing MNIST classificiation.

**Summary Of The Review:**

As it is, the paper does not motivate why their approach should be followed and is not providing a thorough enough evaluation to convince people to follow their approach.

---

### Official Review · Reviewer_qDPG · 2022-10-23

**Confidence:** 4
**Correctness:** 3
**Technical Novelty And Significance:** 2
**Empirical Novelty And Significance:** 1
**Recommendation:** 3

**Clarity, Quality, Novelty And Reproducibility:**

### Clarity
The paper could be significantly improved by addressing grammatical errors, improving the flow, and consistent use of terminology.

- Grammatical errors in the paper slow down readers' understanding. Here are some examples from the first paragraph, with added text in italics: "_the_ impressive success",  "irreplaceable for ~the~ practitioners", "_the_ presence of ~the~ adversarial examples".
- Overall, the flow of the paper could also be improved, with the side effect of making the paper more succinct. Again from the first paragraph, a possible re-write: "Neural networks that are not robust to adversarial examples cannot be deployed in safety-critical tasks such as health care and autonomous driving, since changes in image classification due to signal noise are unacceptable for such tasks."
- Using consistent terminology for algorithm names (e.g. Figure 3 shows "linear activations" and "bounds iterative", while Section 3 refers to "bounds propagation based iterative pruning" and "adversarial preserving layerwise pruning" -- it's not clear to me which is which) would help the reader too.

Some questions and suggestions:
- Q: In figure 1, the red and blue dots are not described in the caption. What are they?

### Quality
The algorithms described in Section 3 correctly maintain the output of the network at either one or two points. Nevertheless, as described in the strengths and weaknesses section, it is unlikely that the robustness estimates provided would be used.

### Novelty
To the best of my knowledge, the pruning techniques presented in this paper are novel.

### Reproducibility
The paper is moderately reproducible. We are unable to evaluate the code since it has not been made available; furthermore, no details on the machine used to run these experiments were provided.

**Strength And Weaknesses:**

The key weakness of the paper is that the estimates computed by running the verification algorithm on local substitutes are neither provably overestimates or underestimates for the original network. Without bounds on how bad the estimates can get (either for individual samples, or on the average estimate for a collection of samples), the estimates would not really be useful for comparing the performance of two different robust networks. It seems possible that a local substitute for a non-robust network would be _more_ robust (e.g. the linear local substitute for a non-robust network).

If the goal is merely to compute an unbiased estimator of the $d_{bnd}$, scaling $d_{adv}$ (which is known to be an overestimate of $d_{bnd}$) could work better.

**Summary Of The Paper:**

The paper introduces two pruning techniques removing non-linearities in a network while preserving local behavior, with an eye towards producing good estimates of the true robust distance $d_{bnd}$ more quickly. The paper empirically evaluates their performance by computing a relative error between "ground truth robust distance" (specifically, lower bounds on the true robust distance, computed using existing verification algorithms) and "estimates on robust distance" (bounds from running the verification algorithms on pruned versions of the original network). The **average** value of the relative error is shown be lower than that where the estimate is obtained using an adversarial attack.

**Summary Of The Review:**

I am recommending rejection of this paper primarily on the basis that improving the relative error to the true robust distance is not useful without a sense for how often this error is an estimate or overestimate. Other aspects of the paper can be improved too, and I've provided more detail (and hopefully constructive suggestions) in the "Clarity, Quality, Novelty and Reproducibility" section.

---

### Official Review · Reviewer_E3L3 · 2022-10-25

**Confidence:** 4
**Correctness:** 1
**Technical Novelty And Significance:** 1
**Empirical Novelty And Significance:** 1
**Recommendation:** 1

**Clarity, Quality, Novelty And Reproducibility:**

Clarity is good.

Quality is poor.

Novelty is poor.

Reproducibility is good.

**Strength And Weaknesses:**

This paper is unfortunately working on a wrong problem. Even if the proposed pruning works perfectly, we still need to prune for each data point and create a different network each time. The over all cost is high and could be higher than simply attacking the original model. It's unclear that we'll save any time or computing resource. On top of this, we'll lose accuracy of the robustness evaluation.

**Summary Of The Paper:**

The paper proposes to prune a neural network for a given data point such that the nearest decision boundary of the pruned network stays approximately the same as the original network. The intended application is robustness evaluation: the proposal is to use the pruned network as a surrogate and thereby save compute.

**Summary Of The Review:**

This paper is unfortunately working on a wrong problem.

---

### Official Review · Reviewer_Eo7V · 2022-10-25

**Confidence:** 5
**Clarity, Quality, Novelty And Reproducibility:** The paper is mostly clear and original.
**Correctness:** 3
**Technical Novelty And Significance:** 2
**Empirical Novelty And Significance:** 1
**Recommendation:** 3

**Strength And Weaknesses:**

### Strength
- It is quite an interesting direction to accelerate robustness evaluation.

### Weakness
- The proposed pruning methods should be directly compared with SOTA robustness-related pruning methods, e.g., Hydra is one of the representatives supporting both adversarial and certifiable robustness [A]. Nothing pruning-related approaches are compared in the paper.
- No SOTA certification methods have been compared. Recently SOTA certification methods have achieved great improvements in scaling to larger models and speeding up certifications. Representative works include the recent VNN-COMP winner alpha-beta-crown, as well as other participants OVAL, VeriNet, Marabou, ERAN, etc (see VNN-COMP21 report [B] for details and VNN-COMP22 for updates). As a certification paper, the paper neither compares with any of them in evaluation nor discusses them in related work.
- To make the results more convincing, it is quite necessary to evaluate the proposed methods on the standard robustness certification benchmarks (e.g., see VNN-COMP or SOTA papers like alpha-beta-crown) such that we can have a good understanding of how much time one can save by using the proposed local substitute networks.
- The proposed pruning methods are local, i.e., pruning the model for each data point. The pruning cost should also be comprehensively discussed in terms of robustness evaluation time.
- Many related works should be mentioned and appreciated in Section 1&2 instead of just citing one SoK paper. Also, the papers considered in related works are quite outdated (mostly before 2019) while many certification papers are proposed in recent years with thousands of certification efficiency improvements.
- Lastly, I am concerned with the certified robustness achieved by the proposed methods since it can only estimate the certified robustness performance without guarantee. However, the sound guarantee is the key factor for certification.

[A] "On Pruning Adversarially Robust Neural Networks", NeurIPS’20

[B] "The Second International Verification of Neural Networks Competition (VNN-COMP 2021): Summary and Results", arXiv’21.

**Summary Of The Paper:**

This paper tries to speed up adversarial/certified robustness assessment by pruning the original large models to smaller sizes. Two different pruning algorithms are proposed targeting certified robustness and adversarial robustness.

**Summary Of The Review:**

Overall, the paper proposes an interesting direction but fails to provide experimental evaluations against SOTA verifiers and robustness-related pruning approaches. Also, a large body of recent works is missing in related works.

---

### Decision · Program_Chairs · 2023-01-20

**Decision:**

Reject

**Justification For Why Not Higher Score:**

The paper provides robustness estimates which are not guaranteed to be related to the actual network robustness. Further the empirical comparisons are quite limited.


**Justification For Why Not Lower Score:**

N/A

**Metareview: Summary, Strengths And Weaknesses:**

While reviewers find the paper's approach interesting and novel, they find the paper lacking on both empirical comparisons and theoretical guarantees. They also find paper lacks a discussion of many related related works.

Authors did not respond to the reviewers comments. I recommend rejection.

Strengths:
1. Novel and interesting approach to efficient robustness evaluation.

Weakness
1. Limited empirical comparisons, missing comparisons with strong baselines.
2. Limited discussion of related results.
3. Lack of robustness guarantees.